

# The unexpected effect of the compound microbial agent NP-M2 on microbial community dynamics in a nonylphenol-contaminated soil: the self-stability of soil ecosystem

Zhaoliang Chen[1,*], Juanqin Zhang[2,*], Weiguang Lv[2,3,4], Hanlin Zhang[2,3,4], Shuangxi Li[2,3,4], Haiyun Zhang[2,3,4], Yue Shen[5], Chunnu Geng[1] and Naling Bai[2,3,4]

[1] School of Ecological Technology and Engineering, Shanghai Institute of Technology, Shanghai, China
[2] Eco-environmental Protection Research Institute, Shanghai Academy of Agricultural Sciences, Shanghai, China
[3] Key Laboratory of Low-carbon Green Agriculture in Southeastern China, Ministry of Agriculture and Rural Affairs, Shanghai, China
[4] Shanghai Key Laboratory of Protected Horticultural Technology, Shanghai, China
[5] Shanghai Agricultural Science and Technology Service Center, Shanghai, China
[*] These authors contributed equally to this work.

Corresponding authors
Chunnu Geng,
gengchunnv@hotmail.com
Naling Bai, bainaling@saas.sh.cn

## ABSTRACT

**Background.** Nonylphenol (NP) is widely recognized as a crucial environmental endocrine-disrupting chemical and persistent toxic substance. The remediation of NP-contaminated sites primarily relies on biological degradation. Compound microbial products, as opposed to pure strains, possess a greater variety of metabolic pathways and can thrive in a wider range of environmental conditions. This characteristic is believed to facilitate the synergistic degradation of pollutants. Limited research has been conducted to thoroughly examine the potential compatibility of compound microbial agents with indigenous microflora, their ability to function effectively in practical environments, their capacity to enhance the dissipation of NP, and their potential to improve soil physicochemical and biological characteristics.

**Methods.** In order to efficiently eliminate NP in contaminated soil in an eco-friendly manner, a simulation study was conducted to investigate the impact of bioaugmentation using the functional compound microbial agent NP-M2 at varying concentrations (50 and 200 mg/L) on the dynamics of the soil microbial community. The treatments were set as follows: sterilized soil with 50 mg/kg NP (CK50) or 200 mg/kg NP (CK200); non-sterilized soil with 50 mg/kg NP (TU50) or 200 mg/kg NP (TU200); non-sterilized soil with the compound microbial agent NP-M2 at 50 mg/kg NP (J50) or 200 mg/kg NP (J200). Full-length 16S rRNA analysis was performed using the PacBio Sequel II platform.

**Results.** Both the indigenous microbes (TU50 and TU200 treatments) and the application of NP-M2 (J50 and J200 treatments) exhibited rapid NP removal, with removal rates ranging from 93% to 99%. The application of NP-M2 further accelerated the degradation rate of NP for a subtle lag period. Although the different treatments had minimal impacts on the soil bacterial $\alpha$-diversity, they significantly altered the $\beta$-diversity and composition of the bacterial community. The dominant phyla were *Proteobacteria* (35.54%–44.14%), *Acidobacteria* (13.55%–17.07%), *Planctomycetes*
(10.78%–11.42%), *Bacteroidetes* (5.60%–10.74%), and *Actinobacteria* (6.44%–8.68%). The core species were *Luteitalea_pratensis*, *Pyrinomonas_methylaliphatogenes*, *Fimbriiglobus_ruber*, *Longimicrobium_terrae*, and *Massilia*_sp003590855. The bacterial community structure and taxon distribution in polluted soils were significantly influenced by the activities of soil catalase, sucrase, and polyphenol oxidase, which were identified as the major environmental factors. Notably, the concentration of NP and, to a lesser extent, the compound microbial agent NP-M2 were found to cause major shifts in the bacterial community. This study highlights the importance of conducting bioremediation experiments in conjunction with microbiome assessment to better understand the impact of bioaugmentation/biostimulation on the potential functions of complex microbial communities present in contaminated soils, which is essential for bioremediation success.

## INTRODUCTION

During the Fourteenth Five-Year Plan period in China, the government emphasized the imperative of enhancing soil pollution prevention and remediation efforts, as well as striving to establish a pristine soil system. The organic pollutants in soil have garnered great attention due to their persistence, accumulation, and "triple" (carcinogenic, teratogenic, and mutagenic) toxicity. Among these pollutants, nonylphenol (NP), which is extensively applied as a crucial raw material and intermediate in fine chemical production, stands out as one of the most concerning environmental endocrine-disrupting chemicals and persistent toxic substances. NP is mainly derived from the decomposition or degradation of nonylphenol polyethoxylates (NPEOs), the second most widely used nonionic surfactant in the world (*Duan et al., 2019*; *Zhang et al., 2023a*). However, NPEOs are recalcitrant to complete mineralization and tend to eventually transform into NP, which is even more toxic and difficult to mineralize than the parent compound. *He et al. (2023)* emphasized that the transformation of NPEO into NP and the subsequent degradation of NP was the rate-limiting processes for controlling the risk of estrogenicity during the biodegradation of NPEO by textile activated sludge. Therefore, since 2021, South Korea has monitored NP and octylphenol as surveillance substances in industrial wastewater treatment plants due to the concerns for human health and aquatic life (*Ryu et al., 2024*). In China, the List of New Key Controlled Pollutants (2023 Edition) clearly states that NP is prohibited as an additive in the production of both pesticide products and NPEOs.

NP can easily accumulate in various organisms as an endocrine-disrupting chemical due to its estrogenic potential, which is approximately $10^{-5}$ to $10^{-4}$ times that of the natural estrogen 17$\beta$-estradiol in binding with the estrogen receptor (*Bhandari et al., 2021*; *Yang et al., 2021*). Both *in vivo* and *in vitro* studies have demonstrated that even at extremely low concentrations, NP is implicated in the occurrence and development of

reproductive, developmental, and immune disorders and various types of cancers (*Dwivedi et al., 2022*; *Guo et al., 2021*). The hydrophobic nature of NP (logKow = 4.48) facilitates its adsorption into soil particles (24-1,059 mg/g), and this process is highly irreversible (*Milinovic et al., 2015*). Therefore, NP residues would significantly affect soil properties and microecological health. For example, the 0.38−6.6 μM NP in agricultural soil induces the germination of the phytopathogen *Fusarium oxysporum*, the stress response of laccase production in *Trametes versicolor*, and negatively affects the structure and function of the soil microbial community (*Kollmann et al., 2003*). *Chen et al. (2020)* reported that under NP stress, earthworms exhibit a decrease in vacuolated glandular cells, thinning of the annulus muscularis, shedding of the cuticle, and thickening of the epidermis. Furthermore, earthworms display an avoidance response to 12.5–200 mg/kg NP; 34% showed inhibited growth at 200 mg/kg NP, and 400 mg/kg NP was strongly lethal to earthworms. The greater the persistence of NPs in soils is, the greater the potential for plant uptake. NP in the soil would decrease the plant quality through food chain bioaccumulation (*Guenther et al., 2002*; *Jiang et al., 2020a*).

The contamination characteristics of NP are not globally uniform (*Zhang et al., 2023a*). In China, the source of NP in the soil is greatly influenced by human activities, such as the application of pesticides containing NPEOs/NPs, wastewater discharge, and sewage sludge returning to the field (*Duan et al., 2021a*; *Jiang et al., 2020b*; *Zheng et al., 2018*). In a soil sample from Heilongjiang Province, the detection rates of NP (max. 156.8 μg/kg), nonylphenol monoethoxylate (max. 15.70 μg/kg), and nonylphenol diethoxylate (max. 13.63 μg/kg) ranged from 68.46% to 95.38% (*Wang et al., 2012*); NP was the least degradable and most likely to accumulate in soil. The application of sludge as an amendment or fertilizer may also pose risks of NP contamination and accumulation in soil (*Duan et al., 2021a*). On this basis, researchers have analyzed NP metabolism *in/ex situ* in various environmental settings, such as soil, water, and sludge (*Hesselsøe et al., 2001*; *Ni et al., 2021*; *Topp & Starratt, 2000*). Biological degradation is mainly responsible for the remediation of NP-contaminated sites (*Wang et al., 2014*). Functional microorganisms, such as *Sphingomonas*, *Sphingobium*, *Pseudomonas*, *Acidovorax*, *Rhodotorula*, *Metarhizium*, and *Chrysosporium*, have been isolated and screened to degrade NP (*Bhandari et al., 2021*; *Duan et al., 2019*; *He et al., 2023*; *Rajendran et al., 2020*; *Yang et al., 2018*). In laboratory hydroponic systems, single strains often demonstrate satisfactory performance, but their effectiveness is often limited in contaminated soils. Conversely, complex microbial products exhibit a wider range of metabolic capabilities and can thrive in diverse environmental conditions, facilitating the synergistic degradation of pollutants (*Liu et al., 2019*). The effective management of persistent organic pollutants requires the cooperation of diverse microbial populations. Numerous studies have confirmed the successful utilization of complex microbial agents for pollutant removal and soil properties promotion. Limited research has been conducted to thoroughly investigate the compatibility of complex microbial agents with indigenous microflora, their stability in soil, their ability to promote NP dissipation, and their potential to enhance soil physicochemical and biological properties. In our previous study, we successfully isolated a highly efficient NP-degrading bacterial consortium, known as NP-M2, which demonstrated the ability to metabolize

75.61% and 89.75% of 1,000 mg/L NP within 48 h and 8 days, respectively (*Bai et al., 2017*). Thus, NP-M2 has been identified as a promising candidate for the bioremediation of NP-polluted environments. However, further investigation is required to determine the feasibility of applying NP-M2 for soil restoration purposes. Consequently, the present study investigated the bioaugmentation of the compound microbial agent NP-M2 by evaluating its degradation efficiency at varying rates of NP. Additionally, the study investigated the dynamics of the soil bacterial community to enhance our understanding of the adaptability of external microbes in contaminated areas and to facilitate the development of strategies aimed at mitigating the adverse impacts of soil NP pollution.

# MATERIALS & METHODS

## Preparation of the compound microbial agent NP-M2

The functional microbial agent NP-M2 exhibits degradation rates of 75.61% and 89.75% for 1,000 mg/L NP within 48 h and 8 d, respectively. The degradation process involves the oxidation of the benzene ring followed by stepwise side-chain biodegradation (*Bai et al., 2017*). The secretion of surface-active compounds by NP-M2 is believed to contribute to the efficient metabolism of NP. Furthermore, NP-M2 primarily consists of *Sphingomonas*, *Pseudomonas*, and *Acidovorax*. Moreover, the single strains of *Sphingomonas*, *Pseudomonas*, and *Acidovorax* obtained from NP-M2 grow well with NP as the sole carbon and energy source after multiple separations and purifications. NP-M2 was cultivated in the minimum salt medium (MSM) (*Bai et al., 2017*) with NP as the sole carbon source at 30 °C and 200 rpm for 8 d. Then, the cells were collected by centrifugation, washed three times with sterilized water, and stored at 4 °C.

## Experimental setup and design

The soil was collected from the topsoil layer (0–20 cm) at Zhuanghang Comprehensive Experimental Station of Shanghai Academy of Agricultural Sciences (30°53′N, 121°23′E), and no NP residue was detected. The soil was identified as sandy loam, with a pH of 6.94, soil organic matter content of 25.96 g/kg, total nitrogen content of 1.32 g/kg, and total phosphorus content of 0.63 g/kg. The soil was sieved through a 2-mm mesh and stored at 4 °C for further analysis.

A total of 24 pots were designed, and 1 kg of soil was loaded into each pot (12 cm high, 14 cm upper radius, 10 cm lower radius) for six treatments. The samples for the six treatments, comprising four replicates each, were categorized into the following groups: CK groups (sterilized soil with 50 and 200 mg/kg NP, respectively); TU groups (non-sterilized soil with 50 and 200 mg/kg NP, respectively); and J groups (non-sterilized soil treated with the compound microbial agent NP-M2 at 50 and 200 mg/kg NP, respectively) (Table 1). NP/methanol was evenly spread onto the surface of the soil and homogeneously mixed, then it was placed in a fume hood overnight until the methanol volatilized completely. Ten milliliters ($10^9$ CFU) of NP-M2 were added to the J50 and J200 treatments, and an equal volume of sterilized water was added to both the CK and TU groups. Then, sterilized water was sprayed to adjust the soil moisture to approximately 25%. Each pot was incubated at 25 °C in the dark for 64 d. Soil samples were nondestructively collected *via* a five-point

**Table 1  Design of different experimental treatments.**

| Items | CK groups | | TU groups | | J groups | |
|---|---|---|---|---|---|---|
| | CK50 | CK200 | TU50 | TU200 | J50 | J200 |
| NP concentration (mg/kg) | 50 | 200 | 50 | 200 | 50 | 200 |
| Soil sterilization | + | + | − | − | − | − |
| NP-M2 addition | − | − | − | − | + | + |

Notes.
CK groups, sterilized soil with the specific content of NP; TU groups, non-sterilized soil with the specific content of NP; J groups, non-sterilized soil with the specific content of NP and the addition of compound microbial agent NP-M2; CK50, sterilized soil with 50 mg/kg NP; CK200, sterilized soil with 200 mg/kg NP; TU50, non-sterilized soil with 50 mg/kg NP; TU200, non-sterilized soil with 200 mg/kg NP; J50, non-sterilized soil with the complex microbial agent NP-M2 addition at 50 mg/kg NP; J200, non-sterilized soil with the complex microbial agent NP-M2 addition at 200 mg/kg NP.
"+/ −" represents the soil sterilization/NP-M2 addition or not, respectively.

sampling method. The samples were sampled at an interval of 8 d up to 32 d and then at 56 d and 64 d, respectively, for a total of 168 samples. The variation in the NP concentration in different treatments was monitored; accordingly, the 24-d samples were selected to analyze the changes in the soil characteristics and microbial community structure.

## Extraction and chromatographic measurement of NP

The collected samples were freeze-dried in a vacuum freeze dryer for subsequent extraction, and the determination of NP was performed according to *Bai et al. (2017)* with slight modifications. The extraction of NP was conducted using an ultrasonic extraction method employing a mixed solution of dichloromethane/methanol (9:1) for 30 min and this process was repeated twice. The resulting organic phase was collected for subsequent vacuum evaporation, while the remaining residues were dissolved in five mL of methanol. Following purification through a silica gel column, the eluent was collected and subjected to nitrogen flow for drying. Finally, a 5-mL methanol solution was employed as the ultimate solvent. The sample was filtered with 0.22 $\mu$m organic filter membrane before high-performance liquid chromatography (HPLC) analysis. An aliquot of the sample solution was automatically injected into the HPLC instrument (Waters 1524; Waters, Milford, MA, USA) equipped with a UV −Vis detector (Waters 2489; Waters, Milford, MA, USA) coupled with a 2707 injector valve. The parameter settings were as follows: An Eclipse C18 column (250 × 4.6 mm × 5 $\mu$m) with 90: 10 methanol: water as the mobile phase, a flow rate of 0.2 mL/min, a detection wavelength of 279 nm, a column temperature of 40 °C, and an injection volume of 20 $\mu$L.

## Determination of soil enzyme activities

The soil enzyme activities were measured according to the methods of *Liu et al. (2020)*. Soil catalase (S-CAT) activity and soil polyphenol oxidase (S-PPO) activity were determined by ammonium molybdate colorimetry and pyrogallol colorimetry, respectively, and hydrogen peroxide and pyrogallic acid were used as the substances. The activity of soil urease (S-UE) and soil sucrase (S-SC) was measured by indophenol blue colorimetry and the 3,5-dinitrosalicylic acid colorimetric method, respectively.

### 16S rRNA gene amplicon sequencing

Genomic DNA was extracted from soil samples using the E.Z.N.A.$^{®}$ Soil DNA Kit (Omega Biotek, Norcross, GA, USA) according to the manufacturer's protocols. DNA quality and quantity were determined using a NanoDrop 2000 Spectrophotometer (Bio-Rad Laboratories Inc., Hercules, CA, USA). Full-length 16S rRNA amplification was performed using the primers 27F (5′-AGRGTTYGATYMTGGCTCAG-3′) and 1492R (5′-RGYTACCTTGTTACGACTT-3′), where the barcode was an eight-base sequence unique to each sample. PCRs were performed in triplicate in a 20 μL mixture containing 4 μL of 5× FastPfu Buffer, 2 μL of 2.5 mM dNTPs, 0.8 μL of each primer (5 μM), 0.4 μL of FastPfu Polymerase, and 10 ng of template DNA. DNA was purified by the Gel Extraction Kit (Axygen Biosciences, Union City, CA, USA) and quantified using QuantiFluor™-ST (Promega, Madison, WI, USA). Then, the purified PCR products were quantified by Qubit$^{®}$3.0 (Life Invitrogen, Carlsbad, CA, USA), and every 24 amplicons whose barcodes were different were mixed equally. The amplicon library was sequenced on a PacBio Sequel II (PacBio, Menlo Park, CA, USA) platform (Shanghai BIOZERON Co., Ltd., Shanghai, China) according to the standard protocols. The raw reads were deposited into the NCBI Sequence Read Archive (SRA) database (accession number: PRJNA1002861).

Operational taxonomic units (OTUs) were clustered with a 98.65% similarity cutoff using UPARSE (version 10, http://drive5.com/uparse/). The phylogenetic affiliation of each 16S rRNA gene sequence was analyzed by the UCLUST algorithm (https://github.com/topics/uclust) against the Silva (SSU138.1) database (http://www.arb-silva.de) using a confidence threshold of 80% (*Kim & Chun, 2014*). Each OTU was annotated to certain phylum, class, order, family, genus, and species. The obtained OTUs were analyzed with respect to $\alpha$ diversity, $\beta$ diversity (principal coordinate analysis, PCoA; partial least squares discriminant analysis, PLS-DA), community composition (bar diagram), and correlation analysis of environmental factors (redundancy analysis, RDA) using the Lingbo MicroClass tools (http://www.cloud.biomicroclass.com/CloudPlatform/home). PICRUST2 and FAPROTAX were used to predict the variation in the predicted functional pathways, and heatmaps were generated using R software (*R Core Team, 2020*).

### Statistical analysis

All the statistical analyses were performed using SPSS (version 25.0, SPSS Inc., Chicago, IL, USA). The line chart and box chart were prepared by Origin (Version 2020; OriginLab Corporation, Northampton, MA, USA). $P < 0.05$ and $P < 0.001$ were considered statistically significant and extremely significant, respectively.

## RESULTS AND DISCUSSION

### NP degradation in different soil treatments

According to the findings presented in Fig. 1, the NP content in the sterilized soil treatments (CK50 and CK200) exhibited a consistent level, with residues remaining above 87.02% until 64 d. The degradation of NP in the various treatments followed a first-order kinetic reaction equation ($R^2 = 0.7848 - 0.9494$). Specifically, at 8 d, the NP degradation efficiencies were 71.35% and 56.96% in the J50 and J200 treatments, respectively, while the TU50 and TU200

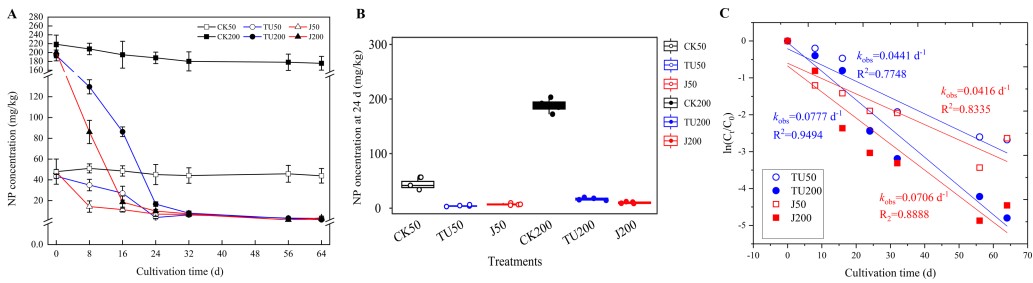

**Figure 1 Concentration curve of residual NP in different treatments (A); statistical analysis of NP content in soil samples at 24 d (B); Biodegradation kinetics analysis (C).** CK50, sterilized soil with 50 mg/kg NP; CK200, sterilized soil with 200 mg/kg NP; TU50, non-sterilized soil with 50 mg/kg NP; TU200, non-sterilized soil with 200 mg/kg NP; J50, non-sterilized soil with the complex microbial agent NP-M2 addition at 50 mg/kg NP; J200, non-sterilized soil with the complex microbial agent NP-M2 addition at 200 mg/kg NP. Each value is mean data ±standard deviation ($n = 4$). "*" and "***" indicate significant differences ($P < 0.05$) and extremely significant differences ($P < 0.001$), respectively.

treatments only achieved degradation efficiencies of 30.00% and 35.23%, respectively. However, during the later stage (24–64 d), the rate of NP removal in the J groups (J50 and J200 treatments) exhibited a significant deceleration, mirroring the trend observed in the TU groups (TU50 and TU200 treatments). Considering the overall trajectory of the TU50 and TU200 treatments, it can be inferred that the soil system possesses a certain degree of self-purification capability. Despite the relatively low initial NP dissipation rate, the TU50 and TU200 treatments ultimately reached 94.05% and 98.21% by 64 d, respectively. This finding suggests the presence of indigenous microorganisms with functional metabolic abilities to degrade NP. This finding aligns with the findings of *Wang et al. (2014)*, who observed that natural attenuation of NP can occur rapidly. Likewise, the degradation rates of the NP-M2 agent treatments (J50 and J200) were 93.10% and 98.98% at 64 d, respectively. Notably, the NP-M2 agent facilitated the removal of NP in the soil within the initial 16 d without a significant period of acclimatization. Consequently, the indigenous microbes in the soil have the capability to degrade NP, and the NP-M2 agent may expedite NP biodegradation.

The degradation pathway by NP-M2 in MSM with NP as the sole carbon source was hypothesized to be initiated by oxidation of the benzene ring, followed by stepwise side-chain biodegradation, with hydroquinone and nonanol as the probable intermediates. However, hydroquinone is difficult to detect due to its instability and susceptibility to oxidation. Similarly, nonanol, an alcohol, is prone to volatilization. Hence, measurements of hydroquinone and nonanol were not included in this study. The decrease in NP degradation rate in 32–64 d may be related to the NP content, NP bioavailability (*Ren et al., 2018*), and the stability and functional characteristics of exogenous microorganisms (*Lourenço et al., 2018*). Bioavailability, which refers to the accessibility of pollutants to soil microbes, significantly impacts bioremediation efficiency, pollutant ecotoxicity, and environmental risk assessment (*Zhang et al., 2023b*). It is comprehensively affected by factors such as soil properties, pollutant characteristics, microbial mobility, and exposure time to contaminants (*Ren et al., 2018*). Lower concentrations of NP in the soil are not

easily accessible to microorganisms, so the degradation rate slows in comparison with that in the active period. Soil carbon fractions also influence the biodegradation of hydrophobic organic contaminants: Unstable hydrolyzable organic carbon (OC) can facilitate pollutant degradation, while alkyl C and arom C-C in stable OC fractions inhibit elimination. The variation dynamics of OC fractions should be explored in future research. Complex environments with biological competition tend to inhibit pollutant degradation by microbial agents since the degradation rate largely depends on the soil internal characteristics and the dynamic changes in the indigenous microflora (*Huang, 2021a*).

## Soil enzyme activities under different treatments

The biodegradation of organic pollutants in soil relies on microbial activity. Soil microbial enzymes (S-CAT, S-UE, S-SC, and S-PPO) were selected to determine the changes in soil microbial activity in different treatments since they play important roles in the mineralization and decomposition of soil organic matter and the circulation and transformation of soil nutrients (*Doran et al., 1994*). S-CAT is an important enzyme that eliminates the toxicity caused by organic pollutants in soil. It can be induced by harmful environmental factors, and the activity reflects environmental stress to a certain degree (*Chabot et al., 2020*). The J50-treated soil exhibited the highest S-CAT activity (48.90 μMol/d/g, $P < 0.05$), and the S-CAT activity decreased in the order of J50>J200>TU200>TU50, indicating that the addition of the NP-M2 agent promoted NP-contaminated soil remediation. NP concentration had a nonsignificant effect on S-CAT activity ($P > 0.05$); NP-M2 affected S-CAT at a dose of 50 mg/kg NP but not at a dose of 200 mg/kg NP (Table 2). S-UE activities ranged from 26.69–30.77 μg/d/g with the highest level observed in the J200 treatment at 30.77 μg/d/g. Nitrification is widely recognized as a sensitive indicator for evaluating the ecotoxicological effects of xenobiotic pollutants in natural ecosystems. For instance, the presence of NP and subsequent biodegradation have been shown to affect the ammonia-oxidizing community (*Wang et al., 2014*). S-SC plays a crucial role in enhancing the solubility of nutrients in the soil, thus acting as a useful indicator for evaluating soil fertility. Compared to the initial S-SC content at 0 d (78.8 mg/d/g), the presence of NP increased the S-SC activity (87.50–143.91 mg/d/g); the highest S-SC was found in the TU200 treatment (143.91 mg/d/g) ($P < 0.05$) (Table 2). Studies have reported that high concentrations of pollutants inhibit S-SC activity (*Cao, 2017*). In the present study, it was speculated that the inhibitory effects of NP might decrease as its content decreases (below 18.70 mg/kg at 24 d); therefore, microbial growth stimulated S-SC activity. S-PPO is a type of redox enzyme that can oxidize aromatic compounds to quinones, promoting the decomposition and transformation of pollutants in soil (*Moradi et al., 2020*). The S-PPO in the TU groups (55.49–62.03 mg/d/g) was significantly higher than that in the J groups (37.41–45.43 mg/d/g) ($P < 0.05$), which may be related to the NP residues in the soil samples. The S-PPO and NP residues were not significantly correlated (Pearson, $P = 0.199$) but showed a similar trend with the S-SC changes. More frequent and dynamic monitoring analyses should be performed to explore the underlying mechanisms involved.

**Table 2** Variation of the soil enzyme activities in different treatments.

| Treatments | S-CAT (μMol/d/g) | S-UE (μg/d/g) | S-SC (mg/d/g) | S-PPO (mg/d/g) |
|---|---|---|---|---|
| TU50 | 47.77 ± 0.30c | 29.54 ± 2.73a | 98.32 ± 4.27b | 55.49 ± 6.43a |
| J50 | 48.90 ± 0.24a | 26.69 ± 2.33a | 87.50 ± 6.23c | 37.41 ± 3.37c |
| TU200 | 47.86 ± 0.66bc | 27.65 ± 2.84a | 143.91 ± 5.68a | 62.03 ± 2.39a |
| J200 | 48.61 ± 0.58ab | 30.77 ± 3.97a | 96.96 ± 5.35b | 45.43 ± 4.91b |

Notes.
TU50, non-sterilized soil with 50 mg/kg NP; J50, non-sterilized soil with the complex microbial agent NP-M2 addition at 50 mg/kg NP; TU200, non-sterilized soil with 200 mg/kg NP; J200, non-sterilized soil with the complex microbial agent NP-M2 addition at 200 mg/kg NP; S-CAT, soil catalase; S-UE, soil urease; S-SC, soil sucrase; S-PPO, soil polyphenol oxidase. Each value consists of mean ± standard deviation ($n = 4$). Different lowercase letters in the same column represent significant levels ($P < 0.05$) reached between treatments.

The biodegradation of organic pollutants is always influenced by microbial activities, and quorum sensing (QS) is a crucial mechanism that regulates physiological activities (*Hoshiko et al., 2022*). Directly adding QS signaling molecules or using bacterial agents that produce such molecules can promote QS regulation, thereby enhancing organic pollutant biodegradation rates. *Sabidi, Hoshiko & Maeda (2022)* reported an increase in methane fermentation when signaling molecules are intentionally added to sewage sludge. Although most studies have shown that strengthening QS regulation is helpful for improving the degradation rate of organic pollutants, some studies have found that QS regulation may negatively impact pollutant degradation. Similarly, pollutants may act as QS inhibitors, which hinder the metabolism of microbial communities (*Hoshiko et al., 2022*). Environmental factors such as pH, temperature, quorum quenching bacteria, and nanoparticles can also negatively impact QS. It might be that NP, or the intrinsic soil properties, affected the QS between NP-M2 and the indigenous microbes; metabolomics analysis would probably be helpful to elucidate this.

## Bacterial community α-diversity and β-diversity analyses

There was no statistically significant difference in the α diversity of the bacterial community, as measured by the Observed species, Chao1, and Shannon indexes, in microcosms contaminated with NP, except for the Pielou index in the J50 treatment (Table 3). The Observed species and Chao1 indexes are used to reflect the richness of the microbial community, that is, the species numbers in the community. The inclusion of NP-M2 agent and different concentrations of NP did not significantly affect the Observed species and Chao1 indexes. The treatment with TU50 exhibited the highest values for Observed species and Chao1, potentially attributable to the stimulation of soil microbial abundance by lower NP concentrations. The Shannon index serves as an indicator of microbial community diversity; no significant disparity ($P > 0.05$) was observed among the four treatments. For the bacterial community evenness, the J50 treatment soil sample had the highest Pielou index value of 0.974 ($P < 0.05$), while there was no significant difference between the TU50, TU200, and J200 treatments. In addition, the Pielou index supported the Shannon estimates that there was a high degree of species richness in the J50 treatment and that the chances of uncertainty with respect to the results would be low.

**Table 3** Analysis of the bacterial community $\alpha$-diversity indexes in different treatments.

| Treatments | Observed species | Chao1 | Shannon | Pielou |
|---|---|---|---|---|
| TU50 | 9,101 ± 707a | 4,028.22 ± 108.93a | 11.29 ± 0.10a | 0.957 ± 0.005b |
| J50 | 8,524 ± 1,990a | 3,573.06 ± 240.97a | 11.42 ± 0.09a | 0.974 ± 0.008a |
| TU200 | 8,831 ± 911a | 3,756.77 ± 190.03a | 11.25 ± 0.16a | 0.958 ± 0.009b |
| J200 | 8,903 ± 702a | 4,016.54 ± 564.95a | 11.21 ± 0.15a | 0.953 ± 0.012b |

Notes.

TU50, non-sterilized soil with 50 mg/kg NP; J50, non-sterilized soil with the complex microbial agent NP-M2 addition at 50 mg/kg; TU200, non-sterilized soil with 200 mg/kg NP; J200, non-sterilized soil with the complex microbial agent NP-M2 addition at 200 mg/kg.

Each value is mean data ± standard deviation ($n = 4$). Different lowercase letters indicate significant differences by LSD test ($P < 0.05$) between treatments.

Compared with the initial bacterial community structure (0 d), the soil bacterial structure at 24 d tended to be clustered (Fig. 2A). At 0 d, the introduction of the NP-M2 agent (J groups) resulted in a significant alteration of the OTU taxon structure compared to that of the TU groups, as evidenced by a 13% shift in PCoA2. The soil samples collected at 24 d clustered together but were distinct from the 0 d samples, as indicated by a 55% separation in PCoA1. This suggested that the cultivation stage had a greater influence on the microbial structure than the concentration of NP or the microbial agent does. Furthermore, with prolonged cultivation time, the external functional microbial agent did not have a noticeable impact on the composition of the indigenous microbes, at least at 24 d. This further verified that the soil system had certain self-stability against external adverse factors (*Luo, 2015*). A subsequent analysis using PLS-DA revealed that the soil bacterial community structure at the OTU level was influenced by different treatments (Fig. 2B). Specifically, P1 (8.15%) distinguished the J50 treatment from the other three treatments, P2 (7.39%) separated the J50 and TU200 treatments from the TU50 and J200 treatments, and P3 (6.91%) distinguished the J200 treatment from the other three treatments. Both the addition of the bacterial agent NP-M2 and the concentration of NP impacted the soil bacterial community structure (PERMANOVA: $P = 0.001$, $P = 0.009$; Table S1). The impact of NP concentration was comparatively more pronounced than that of the NP-M2 addition. Additionally, a reciprocal association between these two factors was observed ($P = 0.001$).

## Microbial community composition analysis

As shown in Fig. 3A, there were 11 taxa with relative abundances above 1% at the phylum level. The dominant groups were *Proteobacteria* (35.54%-44.14%), *Acidobacteria* (13.55%-17.07%), *Planctomycetes* (10.78%-11.42%), *Bacteroidetes* (5.60%-10.74%), and *Actinobacteria* (6.44%−8.68%), with relative abundances above 5%. NP concentrations and NP-M2 agent application did not significantly affect the microbial community composition at the phylum level but influenced their relative abundances to a certain degree (Table 4). The dominant NP-degrading microbes may vary depending on the environment, such as soil type, temperature, oxygen, and reaction medium. For example, *Wu et al. (2022)* reported that *Actinobacteria* (41.59%), beta-*Proteobacteria*, and gamma-*Proteobacteria* are the dominant phyla in the anodic biofilm of a two-chamber microbial fuel cell with

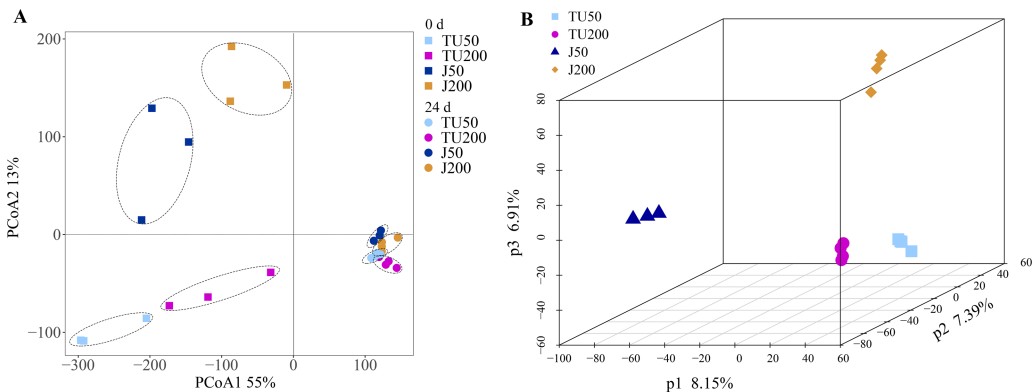

**Figure 2** **PCoA and PLS-DA analyses.** Principal coordinate analysis (PCoA) of the variation in bacterial community structure of different treatments between 0 d and 24 d at OTU level (A); partial least squares discriminant analysis (PLS-DA) of the soil bacterial community structure of different treatments at 24 d at OTU level (B).

an excellent NP degradation performance (82% in 60 h). The relative abundance of the dominant phyla was analyzed in relation to the major effects and interaction effects of NP concentration and NP-M2, as presented in Table 4. Notably, both NP and microbial agent NP-M2 exhibited significant impacts on the relative abundances of *Bacteroidetes* and *Chloroflexi*. Furthermore, an interaction effect was observed specifically on *Chloroflexi* ($P = 0.004$). Notably, *Bacteroidetes* and *Chloroflexi* have previously been recognized for their ability to degrade refractory organics in the environment (*Huang et al., 2021b*). Surprisingly, the J200 treatment displayed the lowest relative abundance of these phyla. Both *Actinobacteria* and *Nitrospirae* were significantly affected by the NP concentration, while the relative abundances of *Proteobacteria* and *Verrucomicrobia* were affected by NP-M2 ($P < 0.05$). In the TU groups, the TU200 treatment enriched the relative abundances of *Proteobacteria* and *Verrucomicrobia* by 24.44% and 39.33%, respectively, compared to the TU50 treatment. The *Verrucomicrobia* phylum was reported to be capable of degrading refractory contaminants (*Li et al., 2022*). *Proteobacteria* are common taxa in the environment and are the leading bacterial group at polluted sites (*Huang et al., 2021b*; *Kabutey et al., 2019*). These differences might be related to the facilitation of NP removal. *Nitrospirae* is a kind of nitrite-oxidizing phylum (*Wang, 2019*), and NP concentration rather than the NP-M2 agent probably affected the nitrogen cycle in soil.

Figure 3B shows the relative abundance of the top 20 genera. The dominant genera (successfully annotated) were *Luteitalea* (7.82%-10.05%), *Lysobacter* (2.62%−5.03%), *Massilia* (2.05%−5.87%), *Pyrinomonas* (2.13%−2.47%), and *Ramlibacter* (2.17%−2.60%), with relative abundances all above 2%. Both the NP concentration and the NP-M2 agent affected the relative abundances of *Sphingomonas* and *Flavisolibacter* ($P < 0.05$) (Table 5). Furthermore, the interaction effect of NP concentration and NP-M2 agent significantly influenced the relative abundances of *Massilia*, *Longimicrobium*, *Steroidobacter*, and *Stenotrophobacter* ($P < 0.05$) (Table 5). Bacterial strains belonging to the genus *Sphingomonas* are frequently obtained from polluted soils owing to their capacity to

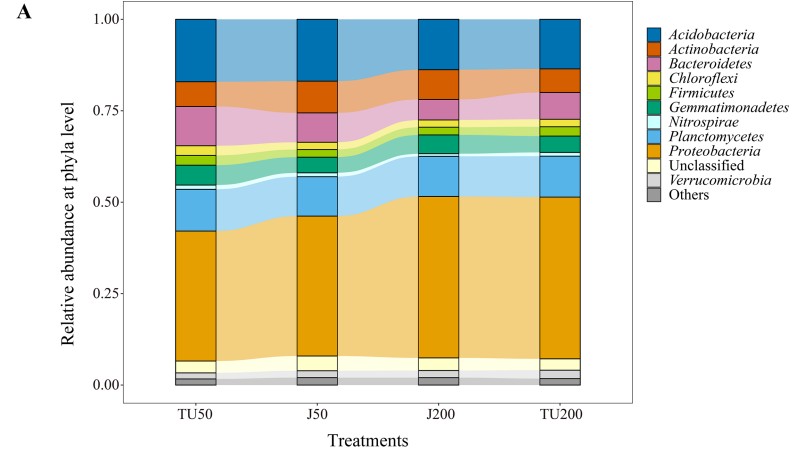

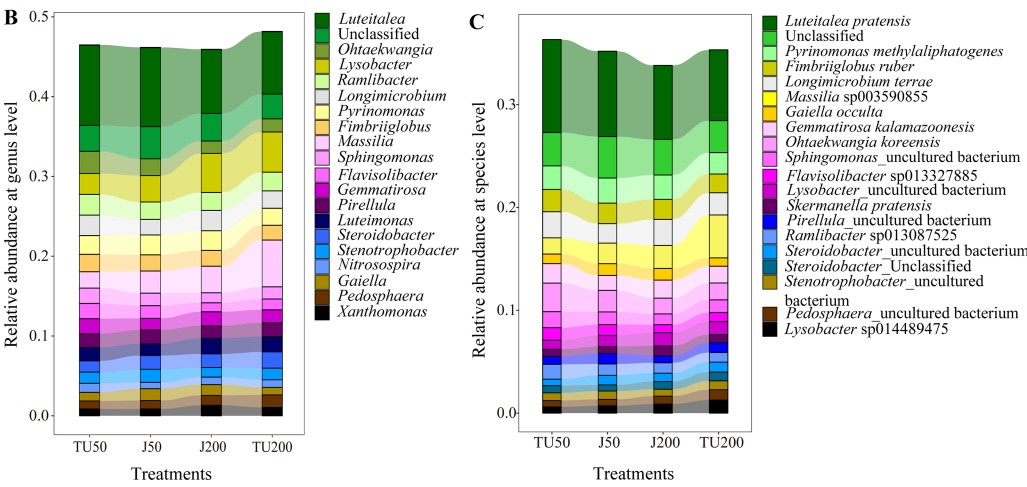

**Figure 3** The relative abundance of the top 11 phyla (A) in the soil samples, and "Others" includes phyla with a relative abundance below 1%; the relative abundances of the top 20 genera (B) and species (C) for the soil samples under different treatments.

metabolize organic pollutants (*Li et al., 2017*; *Hiroshi et al., 2002*). Despite *Sphingomonas* being the prevailing strain in the NP-M2 microbial agent, its enrichment at 24 d was not statistically significant. Likewise, *Pseudomonas* and *Acidovorax* were also not the dominant species in the J groups at 24 d, suggesting the presence of regulatory regularity within the soil system (*Li et al., 2022*). In their study, *Zheng et al. (2018)* documented the prevalence of *Bacillus* as the primary microorganism responsible for the degradation of NP during sludge composting, a finding that contrasts with the outcomes of our investigation. Consequently, the microorganisms involved in the degradation of organic pollutants in the environment can exhibit variability influenced by factors such as oxygen level, reaction medium, temperature, and dissolved organic matter content and composition.

Specifically, *Luteitalea _pratensis*, *Pyrinomonas _methylaliphatogenes*, *Fimbriiglobus _ruber*, *Longimicrobium _terrae*, and *Massilia _sp003590855* were identified as core taxa at the species level, and their presence suggests a potential relationship with

**Table 4  Two-way ANOVA of the relative abundance of dominant bacteria with significant differences at phylum level.**

| Dependent variables | Source | | | | | |
|---|---|---|---|---|---|---|
| | NP concentration | | NP-M2 | | NP concentration*NP-M2 | |
| | F | Significance | F | Significance | F | Significance |
| *Acidobacteria* | 38.810 | 0.000 | 0.003 | 0.955 | 0.133 | 0.722 |
| *Actinobacteria* | 19.397 | 0.001 | 0.506 | 0.492 | 0.006 | 0.940 |
| *Actinobacteria* | 12.946 | 0.004 | 11.736 | 0.006 | 2.253 | 0.161 |
| *Chloroflexi* | 12.627 | 0.005 | 6.258 | 0.029 | 13.307 | 0.004 |
| *Firmicutes* | 4.333 | 0.062 | 0.134 | 0.721 | 0.013 | 0.911 |
| *Gemmatimonadetes* | 0.198 | 0.665 | 0.427 | 0.527 | 4.699 | 0.053 |
| *Nitrospirae* | 5.238 | 0.043 | 1.434 | 0.256 | 0.014 | 0.907 |
| *Planctomycetes* | 0.627 | 0.445 | 0.300 | 0.595 | 1.090 | 0.319 |
| *Proteobacteria* | 2.024 | 0.183 | 18.937 | 0.001 | 3.493 | 0.088 |
| Unclassified | 1.734 | 0.215 | 1.158 | 0.305 | 0.503 | 0.493 |
| *Verrucomicrobia* | 0.132 | 0.723 | 14.646 | 0.003 | 3.098 | 0.106 |

**Notes.**
NP, nonylphenol; NP-M2, the compound microbial agent NP-M2.
The statistical significance of the two-way ANOVA was at $P = 0.05$ level.

NP biodegradation (Fig. 3C). Additionally, the relative abundance of *Sphingomonas _uncultured_bacterium*, belonging to the *Proteobacteria* phylum, was influenced by both the NP concentration and the NP-M2 agent ($P < 0.05$), as indicated in Table 6. The interaction effect of NP concentration and NP-M2 agent significantly influenced the relative abundances of *Longimicrobium _terrae* (belonging to *Gemmatimonadota*), *Massilia _sp003590855* (belonging to *Proteobacteria*), *Luteimonas _sp002324875* (belonging to *Pseudomonadota*), and *Pirellula _staleyi* (belonging to *Planctomycetota*) ($P < 0.05$) (Table 6). The *Luteitalea_pratensis* can mediate extracellular electron transfer to perform oxidation–reduction reactions (*Liu et al., 2023*). *Massilia* has been reported to be beneficial for pollutant biodegradation and soil health (*Chen, Huang & Yuan, 2019*; *Luo, 2015*; *Ruth et al., 2015*). For example, *Luo (2015)* reported that the phenanthrene-degrading bacterium *Massilia* sp. WF1 demonstrated effective performance in contaminated soil, displaying a strong correlation with the rate of phenanthrene degradation. Moreover, the addition of WF1 did not result in significant alterations in the structure of the microbial community pre- and post-treatment. Consequently, the *Massilia* genus may be facilitative in the biodegradation of NP, as evidenced by its highest relative abundance in the TU200 treatment (4.17%).

## Effects of environmental factors on bacterial community structure and functional pathway prediction by the 16S-PICRUSt2 method

Figure 4 shows the effects of environmental factors on soil microbes at the phylum level, with RDA1 and RDA2 explaining 51.68% and 8.32% of the variations, respectively. The ordering axes were generated by explanatory variables (usually environmental factors). RDA1 separated treatments with different NP concentrations (TU50 and J50; TU200

**Table 5  Two-way ANOVA of the relative abundance of the top 20 dominant bacteria at genus level.**

| Dependent variables | Source | | | | | |
| --- | --- | --- | --- | --- | --- | --- |
| | NP concentration | | NP-M2 | | NP concentration*NP-M2 | |
| | F | Significance | F | Significance | F | Significance |
| *Luteitalea* | 21.289 | <0.001 | 0.001 | 0.976 | 0.148 | 0.708 |
| *Lysobacter* | 49.517 | <0.001 | 0.971 | 0.346 | 2.083 | 0.177 |
| Unclassified | 2.345 | 0.154 | 0.898 | 0.364 | 0.345 | 0.569 |
| *Massilia* | 17.371 | 0.002 | 2.950 | 0.114 | 9.655 | 0.010 |
| *Pyrinomonas* | 0.455 | 0.514 | 1.033 | 0.331 | 1.444 | 0.255 |
| *Longimicrobium* | 0.533 | 0.481 | 0.402 | 0.539 | 8.955 | 0.012 |
| *Luteimonas* | 6.501 | 0.027 | 0.331 | 0.577 | 0.331 | 0.577 |
| *Ramlibacter* | 2.407 | 0.149 | 0.110 | 0.747 | 0.393 | 0.544 |
| *Fimbriiglobus* | 0.486 | 0.500 | 1.114 | 0.314 | 0.952 | 0.350 |
| *Steroidobacter* | 2.847 | 0.120 | 0.578 | 0.463 | 7.021 | 0.023 |
| *Ohtaekwangia* | 12.774 | 0.004 | 2.790 | 0.123 | 0.974 | 0.345 |
| *Gaiella* | 3.710 | 0.080 | 0.014 | 0.909 | 0.288 | 0.602 |
| *Gemmatirosa* | 0.135 | 0.720 | 0.406 | 0.537 | 4.548 | 0.056 |
| *Sphingomonas* | 6.056 | 0.032 | 9.541 | 0.010 | 0.159 | 0.698 |
| *Pirellula* | 1.234 | 0.290 | 0.015 | 0.906 | 0.894 | 0.365 |
| *Pedosphaera* | 0.004 | 0.949 | 3.946 | 0.072 | 2.623 | 0.134 |
| *Flavisolibacter* | 12.281 | 0.005 | 6.510 | 0.027 | 0.032 | 0.860 |
| *Xanthomonas* | 4.873 | 0.049 | 0.919 | 0.358 | 0.500 | 0.494 |
| *Stenotrophobacter* | 2.018 | 0.183 | 0.126 | 0.729 | 11.207 | 0.007 |
| *Nitrosospira* | 0.746 | 0.406 | 4.967 | 0.048 | 0.493 | 0.497 |

**Notes.**
NP, nonylphenol; NP-M2, the compound microbial agent NP-M2.
The statistical significance of the two-way ANOVA was at $P = 0.05$ level.

and J200); RDA2 separated treatments with NP-M2 agent addition (J50 and J200) and indigenous microorganisms (TU50 and TU200), indicating significant differences in the bacterial community structure. The variables S-SC, S-CAT, S-PPO, and S-UE exhibited positive correlations with RDA1. S-CAT was negatively correlated with RDA2, whereas S-SC, S-PPO, and S-UE were positively correlated with RDA2. The variables S-SC, S-CAT, and S-PPO exerted significant influences on the structure and distribution of the bacterial communities, except S-UE. The order of influence was as follows: S-PPO>S-SC>S-CAT>S-UE, which corresponded to their respective activities, as outlined in Table 2. Notably, S-PPO may be crucial in the biodegradation of NP. In comparison to soil samples containing 200 mg/kg NP (TU200 and J200 treatments), soil enzymes appeared to have negligible effects on soil samples containing 50 mg/kg NP. *Bacteroidetes* were enriched in the TU50 treatment, while *Actinobacteria* and *Proteobacteria* were enriched in the J50 and J200 treatments, respectively. This tendency was also confirmed by their relative abundances, as shown in Fig. 3A. Similarly, previous research by *Duan et al. (2021b)* reported that *Actinobacteria*

**Table 6  Two-way ANOVA of the relative abundance of the top 20 bacteria at species level.**

| Dependent variables | Source | | | | | |
|---|---|---|---|---|---|---|
| | NP concentration | | NP-M2 | | NP concentration*NP-M2 | |
| | F | Significance | F | Significance | F | Significance |
| *Luteitalea _pratensis* | 10.383 | 0.008 | 0.188 | 0.673 | 1.073 | 0.323 |
| *Ohtaekwangia _koreensis* | 15.603 | 0.002 | 3.390 | 0.093 | 1.804 | 0.206 |
| Unclassified | 0.898 | 0.364 | 2.345 | 0.154 | 0.345 | 0.569 |
| *Longimicrobium _terrae* | 0.713 | 0.416 | 0.757 | 0.403 | 10.653 | 0.008 |
| *Pyrinomonas _methylaliphatogenes* | 1.155 | 0.306 | 2.988 | 0.112 | 0.297 | 0.597 |
| *Fimbriiglobus _ruber* | 6.035 | 0.032 | 0.100 | 0.758 | 3.328 | 0.095 |
| *Massilia _sp003590855* | 11.136 | 0.007 | 3.193 | 0.102 | 7.727 | 0.018 |
| *Gemmatirosa _kalamazoonesis* | 0.018 | 0.897 | 1.339 | 0.272 | 4.160 | 0.066 |
| *Ramlibacter _sp013087525* | 4.504 | 0.057 | 0.876 | 0.369 | 2.144 | 0.171 |
| *Sphingomonas _uncultured_bacterium* | 4.864 | 0.050 | 4.967 | 0.048 | 0.432 | 0.524 |
| *Flavisolibacter _sp013327885* | 21.984 | <0.001 | 3.635 | 0.083 | 0.151 | 0.705 |
| *Lysobacter _uncultured_bacterium* | 6.259 | 0.029 | 0.689 | 0.424 | 0.584 | 0.461 |
| *Luteimonas _sp002324875* | 0.419 | 0.531 | 5.820 | 0.034 | 7.648 | 0.018 |
| *Gaiella _occulta* | 0.377 | 0.552 | 6.698 | 0.025 | 0.276 | 0.610 |
| *Pirellula _staleyi* | 0.125 | 0.730 | 0.939 | 0.353 | 6.143 | 0.031 |
| *Steroidobacter _Unclassified* | 5.874 | 0.034 | 2.588 | 0.136 | 0.000 | 0.984 |
| *Stenotrophobacter _uncultured_bacterium* | 0.010 | 0.924 | 0.419 | 0.531 | 1.526 | 0.242 |
| *Xanthomonas _albilineans* | 3.094 | 0.106 | 1.449 | 0.254 | 1.295 | 0.279 |
| *Pedosphaera _uncultured_bacterium* | 10.164 | 0.009 | 2.113 | 0.174 | 2.207 | 0.165 |
| *Nitrosospira _sp900142705* | 0.013 | 0.911 | 1.296 | 0.279 | 0.506 | 0.492 |

**Notes.**
NP, nonylphenol; NP-M2, the compound microbial agent NP-M2.
The statistically significance of the two-way ANOVA was at $P = 0.05$ level.

and *Proteobacteria* phyla are core flora involved in NP removal during sludge anaerobic digestion.

PICRUST2 functionally predicted pollutant degradation pathways in soil samples with different NP concentrations and NP-M2 additions (Fig. 5). When NP was applied at 200 mg/kg (J200 and TU200 treatments), metabolism (*e.g.*, metabolism of other amino acids, xenobiotics biodegradation and metabolism), environmental information (*e.g.*, signal transduction, membrane transduction), and cellular processes (cell mobility) functions were enriched. The activation of enzymes or QS may be responsible for the response of these signals to high-concentration NP metabolism. In addition, the J50 and TU50 treatments exhibited enrichment in basic metabolisms, such as nucleotide metabolism, amino acid metabolism, and carbohydrate metabolism. This suggested that bacteria can activate alternative metabolic pathways in response to adverse environmental stimulation. The soil system can self-restore and effectively manage external adverse changes, potentially surpassing the efficacy of the bioaugmentation method in the long term. Similarly, genes involved in aspects of the bisphenol A (another well-known endocrine disrupting compound often detected with NP/NPEOs)-resistance response, such as base excision repair, osmoprotectant transport, iron-complex transport, and some energy

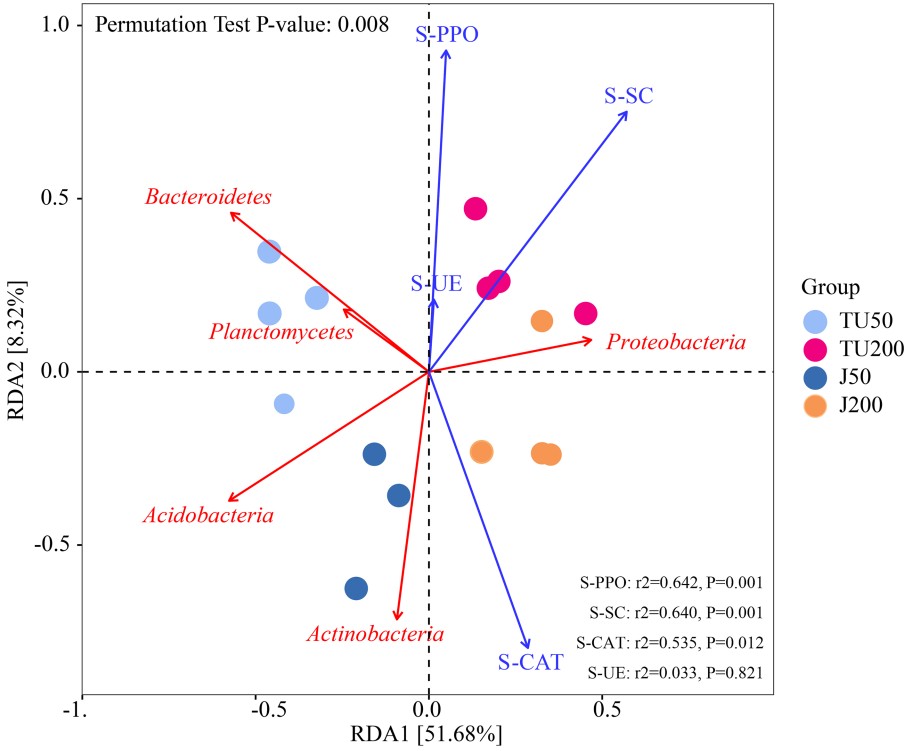

**Figure 4** Redundancy analysis (RDA) of the correlations between environmental factors, bacterial phyla, and bacterial community composition under different treatments.

metabolisms, were upregulated to mitigate the loss of energy associated with the exposure (*Tian et al., 2023*). More importantly, major shifts in bacterial function were caused by the NP concentration and, to a lesser extent, by the microbial agent NP-M2 (Fig. 5).

Ammonia-oxidizing archaea are more responsive to NP application than ammonia-oxidizing bacteria are (*Wang et al., 2014*). In this study, FAPROTAX prediction revealed that the J groups and TU groups were separated and showed different tendencies, indicating that both the NP-M2 agent and the NP concentration affected the nitrogen cycle. The J groups exhibited an increase in nitrate respiration, nitrogen respiration, and nitrate reduction, while the TU groups showed an increase in aerobic nitrite oxidation, aerobic ammonia oxidation, and nitrification (Fig. S1). The relative abundance of ureolysis function (Fig. S1) did not show a positive correlation with S-UE activity (Table 2). Therefore, further investigation is needed to thoroughly explore the underlying relationship between NP biodegradation and the nitrogen cycle. *Toole et al. (2021)* reported that the gene frequencies in certain pathways, particularly those related to soil carbon and nitrogen cycling, were significantly underestimated by PICRUSt2. To obtain a more comprehensive understanding of bacterial functions under different treatments, it is suggested to combine PICRUSt2 with FAPROTAX. Furthermore, the integration of DNA stable-isotope probing, metagenomics analysis, and culture-dependent methods for functional validation enables a

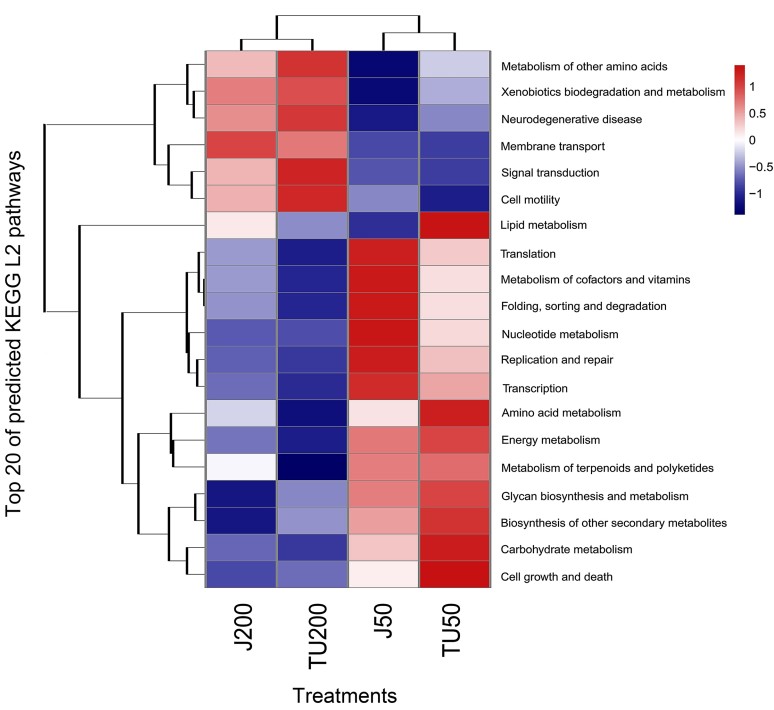

**Figure 5  Heatmap of KEGG L2 functional pathway of the soil samples predicted by PICRUSt2 under different treatments.**

more precise and in-depth analysis of microbiological processes in complex environmental samples.

## CONCLUSIONS

The indigenous microorganisms (TU50 and TU200 treatments) and the application of microbial agent (J50 and J200 treatments) demonstrated high efficiency in the removal of NP in soil, with removal rates ranging from 93% to 99%. The addition of microbial agents resulted in an accelerated degradation rate of NP without a significant lag period. The dominant phyla observed were *Proteobacteria* (35.54%–44.14%), *Acidobacteria* (13.55%–17.07%), *Planctomycetes* (10.78%–11.42%), *Bacteroidetes* (5.60%–10.74%), and *Actinobacteria* (6.44%−8.68%), all of which had relative abundances exceeding 5%. The core taxa identified were *Luteitalea _pratensis*, *Pyrinomonas _methylaliphatogenes*, *Fimbriiglobus _ruber*, *Longimicrobium _terrae*, and *Massilia _*sp003590855. The bacterial community structure and taxon distribution were notably influenced by the major environmental factors, namely, S-CAT, S-SC, and S-PPO. The fluctuations in the bacterial community structure and functional pathways were more pronounced in response to changes in the NP concentration than the application of the NP-M2 microbial agent. The soil ecosystem exhibited a certain level of biological self-stability to maintain its equilibrium. In summary, the microbial agent NP-M2 efficiently degraded NP in laboratory settings, but its stability and management in soil systems require further investigation (*e.g.*, DNA

stable-isotope probing, fluorescence labeling, metabolomics analysis, and metagenomics analysis).

## ACKNOWLEDGEMENTS

The authors are grateful to the anonymous reviewers for their constructive comments in improving both the language and scientific quality of the manuscript.

### Funding

This work was supported by the Shanghai Natural Science Foundation of the Shanghai Science and Technology Commission (21ZR1464500), the Shanghai Agriculture Applied Technology Development Program, China (2022-02-08-00-12-F01152), and the Outstanding Team Program of the Shanghai Academy of Agricultural Sciences (Hu-Nong-Ke-Zhuo 2022 (008)). The funders had no role in study design, data collection and analysis, decision to publish, or preparation of the manuscript.

### Grant Disclosures

The following grant information was disclosed by the authors:
Shanghai Natural Science Foundation of the Shanghai Science and Technology Commission: 21ZR1464500.
Shanghai Agriculture Applied Technology Development Program, China: 2022-02-08-00-12-F01152.
Outstanding Team Program of the Shanghai Academy of Agricultural Sciences: Hu-Nong-Ke-Zhuo 2022 (008).

### Competing Interests

The authors declare there are no competing interests.

### Author Contributions

- Zhaoliang Chen performed the experiments, analyzed the data, prepared figures and/or tables, and approved the final draft.
- Juanqin Zhang performed the experiments, analyzed the data, authored or reviewed drafts of the article, and approved the final draft.
- Weiguang Lv conceived and designed the experiments, prepared figures and/or tables, authored or reviewed drafts of the article, and approved the final draft.
- Hanlin Zhang analyzed the data, prepared figures and/or tables, and approved the final draft.
- Shuangxi Li conceived and designed the experiments, prepared figures and/or tables, authored or reviewed drafts of the article, and approved the final draft.
- Haiyun Zhang analyzed the data, prepared figures and/or tables, and approved the final draft.
- Yue Shen performed the experiments, authored or reviewed drafts of the article, and approved the final draft.

- Chunnu Geng conceived and designed the experiments, authored or reviewed drafts of the article, and approved the final draft.
- Naling Bai conceived and designed the experiments, performed the experiments, analyzed the data, authored or reviewed drafts of the article, and approved the final draft.

## Data Availability

The data is available at figshare: Bai, Naling (2023). The bacterial community dynamics in a nonylphenol-contaminated soil with/without functional microbial agent NP-M2 addition. figshare. Dataset. https://doi.org/10.6084/m9.figshare.24407956.v2.

## Supplemental Information

Supplemental information for this article can be found online at http://dx.doi.org/10.7717/peerj.17424#supplemental-information.

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
