# Peer review of "The unexpected effect of the compound microbial agent NP-M2 on microbial community dynamics in a nonylphenol-contaminated soil: the self-stability of soil ecosystem"

_PeerJ, doi:10.7717/peerj.17424_

## Round 0.1 · original submission · Major Revisions

Please answer the concerns of both reviewers.

**Language Note:** The review process has identified that the English language must be improved. PeerJ can provide language editing services - please contact us at [email protected] for pricing (be sure to provide your manuscript number and title). Alternatively, you should make your own arrangements to improve the language quality and provide details in your response letter. – PeerJ Staff

Reviewer 1 ·

Basic reporting

This paper "The unexpected effect of compound microbial agent NP-M2 on microbial community dynamics in a nonylphenol-contaminated soil: the self-stability of soil ecosystem" assesses the remotion of NP in soil by microbial community, using full-length 16S sequencing. This results offers valuable insights, but it's important to note that the study only examined the NP reduction, changes in microbial community and some enzyme activities. However, there are specific concerns outlined.

1.1 The enhancement to the English language are necessary to facilitate better understanding for a global readership.

1.2 I suggest that figures and tables be well labeled and described, including statistics, number of replicates, and complete information.

Experimental design

2.1 The methods for determining enzymatic activity lack information and reference of the methods used.

2.2 Why did you used the SILVA 138 database for bacteria taxonomy?

Validity of the findings

3.1 Given the difficulty in attaining a normal distribution for bacterial data, it might be more suitable to explore non-parametric analysis or a two-way ANOVA, particularly considering the involvement of two variables in this study.

Additional comments

Line 71: In vivo and in vitro should be italicized.
Line 90: References should be arranged alphabetically.
Line 96: In/ex situ should be italicized.
Line 119: Should be given more information of "microflora NP-M2".
Line 162: You should modify the term "soil microbiological analyses", or separate "enzyme activities" and "microbiota identification".
Line 179: The reference (Callahan et al.) is incomplete.
Line 262 - : Bacterial phyla are not italicized.

Reviewer 2 ·

Basic reporting

There are many parts of the text that are too long for a single sentence to convey. It may be beneficial to consider revising the text by seeking assistance from an English proofreader. The article effectively presents its arguments, providing clear and concise information on the introduction, background, and significance of the authors' research. Nevertheless, it employs some technical abbreviations and jargon specific to its field, which may hinder understanding for readers from other disciplines. Therefore, the clarity and comprehensibility of this publication could benefit from editing measures aimed at enhancing its accessibility to a broad audience of readers. The journal's guidelines advise spelling out uncommon abbreviations at their first occurrence to avoid confusion. There are additional areas where the standards are not adhered to. The directions should be followed in full. Secondly, the image quality of the figures was poor, with some lacking axis labels. Furthermore, the explanations of the figures were inadequate, rendering it difficult to interpret the results without reference to the main text. It is important to ensure that the figures provide sufficient information for researchers in the same field, even without reference to the main text.

Experimental design

This article presents an innovative exploration of the practical application of previously formulated biological agents. Thus, it convincingly demonstrates alignment with the journal's scope. However, certain experimental methods employed in the document are not clearly outlined, creating the need for further scrutiny.

Validity of the findings

The research's importance and background are clear. However, the conclusions are ambiguous regarding what can be inferred from the findings and what they are advocating for.

Additional comments

[Abstract]
L37-38: Unique abbreviations such as CK, TU, J, etc. should not be used when grouping.
L43: Decimal points such as 93.10%-98.98% are redundant.
L46: Abbreviations have suddenly appeared. Abbreviations that are not commonly used in various fields should also be avoided.

[Introduction]
L69-71: Reconsider the statement. The text is difficult to understand.
L71: “In vivo” and “in vitro” should be italicized.
L81-86: Reconsider the statement. Sentences are too long and poorly connected.
L82: Molar should be corrected to micromolar.
L91-92: Is the detection limit (ND) necessary? Could it be presented as the maximum value, such as max. xxx µg/kg?
L91: There is no explanation of the abbreviations NP1EO and NP2EO.
L114: Either the “16S rDNA” or “16S rRNA” gene should be consistently used throughout the entire text.
L140: Ten milliliters of NP-M2 (109 CFU).
L162-L167: It only presents a roster of the experimental techniques.
L175-176: Needs more context to improve the text. At least a clear description of what the library prep kit was used is and how the barcode was added to each library is necessary.
L179: No explanation of the conditions utilized in DADA2 (e.g., similarity) has been provided.
L181: Why are you applying not only DADA2 but also DADA on this step?
L168-186: No explanation has been provided for the use of PICRUSt2 or any other methods of analysis. The purpose for which these tools are being employed should be clearly stated.
L187-191: It should also explain how it was diagrammed.

[Results and Discussion]
Overall, the text requires extensive scrutiny of its significant figures and use of abbreviations, as it currently presents difficulties for the reader.
L193: A discussion on the degradation products of NP is necessary.
L204-L207: Reaction kinetics should also be examined.
L220-223: Reconsider the statement. The text is lengthy and contains unfamiliar abbreviations.
L236: A discussion on quorum sensing is necessary, specifically regarding the correlation between degradation speed and enzyme activity. Sabidi et al. (doi: 10.1007/s00253-022-12014-w) have reported an increase in methane fermentation when signaling molecules are intentionally added to sewage sludge. Also, pollutants have been observed to act as quorum sensing inhibitors, which hinder the metabolism of microbial community (doi: 10.1016/j.chemosphere.2022.134253.).
L239-245: Reconsider the statement. The text is lengthy.
L258: Given the artificial introduction of biological agents in this scenario, it becomes imperative to delve into absolute numerical comparisons per unit when assessing bacterial communities.
L265-268: If this biological agent isn't specifically designed for use solely in China, it would be prudent to discuss its potential benefits when employed in diverse soil conditions across different countries.
L258 & L310: How about a configuration as described below? To present a comprehensive overview, you'll begin by elucidating the broader perspective of the bacterial community, discussing both alpha and beta diversity metrics. Subsequently, you'll shift your focus to examine the distinct bacterial species identified within each sample, providing a more detailed analysis of their composition and characteristics.
L355 & L356: A brief explanation should be given of what RDA1 and RDA2 mean.
L355-359: There should be a clear description of what is good about this result.
L361-362: Which particular signals undergo alteration? Do these signals correspond to each enzyme activity? If they do, what implications arise from these changes?
L364-369: Reconsider the statement. Sentences are too long.
L371: BPA has emerged abruptly; it's important to provide a succinct explanation of the relationship between NP and BPA.
L381-382: PICRUSt2 is an estimate of the expected change in function due to changes in the bacterial community and does not reflect gene expression. Therefore, this tool should not be dismissed, but rather discussed in terms of changes in the proportion of specific genes present.
L383: The presented findings appear to be a compilation of results rather than a comprehensive conclusion.
L393-395: It does not clearly state what specific measures are needed during the discussion.

---

## Round 0.2 · Minor Revisions

Please address the last comment of the reviewer 2

Reviewer 1 ·

Basic reporting

No comment

Experimental design

No comment

Validity of the findings

No comment

Additional comments

Successfully addressed the changes suggested to improve the writing and quality of the article.

Reviewer 2 ·

Basic reporting

The manuscript has been significantly improved, making it a more reader-friendly article. Please make one improvement.
The visibility of the figures has been generally enhanced. However, the characters in Figure 3, especially at the species level, are unclear. Please consider making the text clearer, such as by enlarging the diagram.

Experimental design

No comment.

Validity of the findings

No comment.

---

## Round 0.3 · accepted · Accept

Thanks for addressing all the reviewers comments.